# Mucin Transiently Sustains Coronavirus Infectivity through Heterogenous Changes in Phase Morphology of Evaporating Aerosol

**DOI:** 10.3390/v14091856

**Published:** 2022-08-24

**Authors:** Robert W. Alexander, Jianghan Tian, Allen E. Haddrell, Henry P. Oswin, Edward Neal, Daniel A. Hardy, Mara Otero-Fernandez, Jamie F. S. Mann, Tristan A. Cogan, Adam Finn, Andrew D. Davidson, Darryl J. Hill, Jonathan P. Reid

**Affiliations:** 1School of Cellular and Molecular Medicine, University of Bristol, Bristol BS8 1TD, UK; 2School of Chemistry, Cantock’s Close, University of Bristol, Bristol BS8 1TS, UK; 3Bristol Veterinary School, University of Bristol, Langford House, Langford, Bristol BS40 5DU, UK

**Keywords:** bioaerosol, COVID-19, airborne disease transmission, mucus, CELEBS

## Abstract

Respiratory pathogens can be spread though the transmission of aerosolised expiratory secretions in the form of droplets or particulates. Understanding the fundamental aerosol parameters that govern how such pathogens survive whilst airborne is essential to understanding and developing methods of restricting their dissemination. Pathogen viability measurements made using Controlled Electrodynamic Levitation and Extraction of Bioaerosol onto Substrate (CELEBS) in tandem with a comparative kinetics electrodynamic balance (CKEDB) measurements allow for a direct comparison between viral viability and evaporation kinetics of the aerosol with a time resolution of seconds. Here, we report the airborne survival of mouse hepatitis virus (MHV) and determine a comparable loss of infectivity in the aerosol phase to our previous observations of severe acute respiratory syndrome coronavirus 2 (SARS-CoV-2). Through the addition of clinically relevant concentrations of mucin to the bioaerosol, there is a transient mitigation of the loss of viral infectivity at 40% RH. Increased concentrations of mucin promoted heterogenous phase change during aerosol evaporation, characterised as the formation of inclusions within the host droplet. This research demonstrates the role of mucus in the aerosol phase and its influence on short-term airborne viral stability.

## 1. Introduction

Evidence indicates that the coronavirus causing COVID-19, severe acute respiratory syndrome coronavirus 2 (SARS-CoV-2), can be rapidly spread though the transmission of aerosolised expiratory secretions in the form of droplets or particulates [1,2,3]. The physicochemical properties of exhaled aerosol, and their transformation in environmental conditions, impact on the airborne survival of pathogens contained in the droplets and may impact on the use of mitigations to restrict their spread.

Previous research suggests that the evaporative mass flux of water from aerosol containing biological organisms (referred to as bioaerosol below) can dictate ultimate microbial viability outcomes [4]. Oswin et al. demonstrated that the stability of an airborne coronavirus (mouse hepatitis virus; MHV) is dependent on environmental conditions including relative humidity (RH) and temperature [5]. RH and temperature govern transient and equilibrium changes in bioaerosol microphysics. For example, the initial water activity of an exhaled droplet is higher than that of the environmental RH. As a result, this difference leads to mass and heat flux from the droplet through the evaporation of water until the remaining moisture content is in equilibrium with the environmental RH [6]. The evaporation of water from the bioaerosol results in an increase in concentration of solutes in the aerosol, such as salt, protein, and organic material. Indeed, the supersaturation of solutes at low environmental RH can result in the crystallisation of the salt fraction of the bioaerosol in a phenomenon known as efflorescence. The evaporation of a bioaerosol particle to the efflorescence point of dissolved salts results in a particle containing solid crystals and a viscous organic phase. There is evidence for a near instantaneous loss of viral infectivity as a result of bioaerosol efflorescence, although the direct mechanisms behind the loss of infectivity are unknown [5]. Some studies have suggested that droplet efflorescence can protect virions [7,8] within the salt crystal formed on crystallisation. Given the uncertainty in the underlying mechanisms that determine the loss of viral infectivity in aerosol, there is a need for the role of the particle phase in influencing viral infectivity in aerosol to be explored in much greater detail, and on the time scales relevant to phase changes within particles.

Respiratory bioaerosol droplets can range from <100 nm to >500 µm diameter [9,10] and, cumulatively across this size distribution, have a large surface area-to-volume ratio when compared to an equivalent volume of bulk liquid. Following exhalation, the low environmental CO_2_ concentration (when compared to the high CO_2_ concentration in the lungs) can be expected to lead to an irreversible partitioning of dissolved CO_2_ from the droplet into the gas phase [11,12]. Evaporation of CO_2_ leads to an adjustment in the solution phase equilibria with a decrease in bicarbonate anion and H^+^ ion concentrations, and an increase in the droplet pH. Although the effects of pH on coronavirus stability have been explored in the bulk phase [12,13], the detrimental contribution of increased bioaerosol pH to airborne virus infectivity requires further understanding. 

Considering the physicochemical processing of respiratory aerosol on exhalation, it is critical to consider the influence of the molecular complexity of respiratory secretions in the study of infectious aerosol. One major component of respiratory secretions commonly hypothesized to affect viral infectivity is mucus [14,15,16], a complex mixture of glycoproteins forming a hydrogel with water, other cellular debris, and protective factors [17]. In vivo mucus acts as both a physical barrier against infection from pathogens and contains mediators of immunity such as immunoglobins and lysozyme [18]. Upregulation of the building blocks of secretory airway mucus, gel forming mucin 5AC and mucin 5B (MUC5AC, MUC5B), is a common symptom of both respiratory and oral disease state [19,20,21,22]. Equally, soluble mucin 7 (MUC7) found in saliva has antimicrobial properties but lacks gel forming properties [23,24]. Despite the importance of mucins in airway and oral physiology, an understanding of the role of mucin in airborne disease transmission is limited. In an exhaled aerosol, there is potential for mucin to affect viral infectivity directly (e.g., through binding the viral particle), indirectly (e.g., through altering the evaporation dynamics of the droplet itself), or a combination of both.

Reported here is an exploration of how mucin affects the infectivity of a coronavirus (MHV) in the aerosol phase. This is accomplished through a representative model of respiratory bioaerosol, specifically aerosolized MHV in a mammalian basal cell culture. The working stocks were spiked with clinically relevant concentrations of type II porcine gastric mucin. Type II gastric mucin contains mucin 2 and mucin 6 (MUC2/MUC6), as well MUC5AC [24,25,26], and although these are not the major secreted mucins in saliva [27,28], they are gel-forming and serve as a surrogate for the secretory gel-forming mucins in saliva upregulated during respiratory disease. Type II gastric mucin is an established component of simulated saliva fluid [29] and has been used in aerosol and bulk liquid viral infectivity studies [30,31,32]. Thus, they provide a route to exploratory studies of the influence of mucin composition on aerosol physicochemical transformations and viral infectivity over timescales not previously accessed by laboratory measurements.

Suspending bioaerosol in an electrodynamic field, using the technique Controlled Electrodynamic Levitation and Extraction of Bioaerosol onto Substrate (CELEBS), allows for the reproducible measurement of viral viability [33]. The Comparative Kinetics Electrodynamic Balance (CKEDB) uses similar technology to levitate single particles for aerosol microphysics measurements. The trapped particles are illuminated via a laser, which allows particle radius and phase to be inferred from the angularly resolved light scattering patterns [34]. Combined, these two approaches provide insight into the underlying physicochemical properties and evaporation dynamics that drive the loss in viral infectivity in response to bioaerosol size, relative humidity (RH), and mucin concentration.

## 2. Materials and Methods

### 2.1. Cell Culture Methods 

Mouse fibroblast cells (17-CL1) [35] were cultured at 37 °C and 5% CO_2_ in Dulbecco’s Modified Eagle’s Medium (DMEM, high glucose; Sigma-Aldrich, St. Louis, MO, USA) supplemented with 10% *v*/*v* foetal bovine serum (FBS, Sigma-Aldrich), 100 units·mL^−1^ penicillin (Gibco, Thermo Fisher Scientific, Waltham, MA, USA), 100 µg·mL^−1^ streptomycin (Gibco) and 1% *v*/*v* L-glutamine (Gibco).

### 2.2. Virus Propagation and Titration 

Working viral stocks of MHV-A59 [36] were cultured using the protocol of Leibowitz et al. [37]. The virus titre was determined by an end-point dilution assay, whereby a ten-fold serial dilution of working virus stock was set up and inoculated onto 17-CL1 cells in a 96-well plate (Sigma-Aldrich). After a two-day infection period, the wells were checked for any sign of cytopathic effect and the dilution at which 50% of wells were infected was used to calculate the tissue culture infectious dose 50 (TCID_50_). The TCID_50_ was used to calculate the infectious units per mL (u·mL^−1^) using the Reed and Muench method [38]. The propagation of the virus for subsequent experiments was optimized to achieve an increased number of infectious (i.e., biologically active) units per droplet. Here, a stock harvested 24 h post-inoculation gave the highest yield with 4.2 × 10^7^ u·mL^−1^, Figure 1a. The increased infectious titre translated to an increase in infectious units per droplet between the stock harvested 24 h post-inoculation compared with that harvested at 72 h post-inoculation, Figure 1b.

### 2.3. Airborne Viral Stability Measurements Using CELEBS 

To determine the short-term effects on aerosolised virus infectivity, aliquots of the MHV working stock were aerosolised and levitated within the CELEBS instrument [33]. A square wave form was applied to a droplet-on-demand dispenser (DoD, Microfab Technologies, Plano, TX, USA) to generate individual bioaerosol droplets. The initial bioaerosol droplet volume is a product of the square wave form pulse width and voltage applied to the dispenser, as described by Oswin et al. [5]. A pulse width of 30 µs and voltage of 40 to 45 V corresponds to an initial bioaerosol droplet radius of 25 ± 1 µm, whereas a pulse width of 120 µs and a voltage of 60 to 63 V corresponds with an initial bioaerosol radius of 31 ± 1 µm. Droplets were dispensed into the CELEBS chamber and caught within an electrodynamic trap, Appendix A. After a desired levitation period, the droplet or particles were collected into 5 mL of DMEM (10% FBS) and analysed using an infectivity assay [5]. The collection substrate was transferred evenly across a 96-well plate containing MHV target 17-CL1 cells. After a 48-h incubation period, wells were observed for cytopathic effect, where viral infectivity and virus per droplet were calculated using a Poisson distribution [38]. Levitations (5 s) at 90% RH were shown to result in no observable loss of infectivity compared with the initial viral titre measured through TCID_50_ [12]. A full list of the viability measurements is included in the Appendix A. The data from each separate measurement requires standardisation to an average of the initial viral load carried by a droplet at the beginning of the levitation, referred to as the T = 0 measurement from that day. The standardised distribution of T = 0 has been presented in Appendix A. The remaining infectivity recorded from a single levitation of a population of particles is referred to as one measurement. Infectivity was inferred following at least three measurements, i.e., measurements with at least three populations of particles, with each population containing between 3 and 13 particles. The data presented are the mean of N populations. Unless stated otherwise, error bars show standard error. 

To measure the effects of RH on virus viability and compare this with similar infectivity studies with SARS-CoV-2, bioaerosol droplets were levitated at 40% and 90% RH. RH was controlled through a mixing ratio of humidified and dry air flow (200 mL·min^−^^1^). Temperature was maintained at 19 °C by Peltier cooling of a copper block at the inflow of gas supply to the chamber. RH and temperature measured continuously by a HIH-4062-C (Honeywell, Charlotte, NC, USA) temperature and RH sensor.

To determine if initial bioaerosol droplet radius influences viral infectivity, two droplet volumes were levitated for five minutes at 40% RH. Constrained by the limits of the dispenser, a 1.8-fold increase in initial aerosol volume was probed, comparing droplets of a 25.5 ± 1 with 31 ± 1 µm initial radius.

Finally, to investigate the effects of mucin concentrations on viral infectivity, CELEBS measurements were carried out using DMEM (10% FBS) spiked with mucin (porcine stomach type II, pH 6.5, Sigma-Aldrich), where the amount added to the viral working stock was at clinically significant concentrations (0.1, 0.3 and 0.5% *w*/*v*) [21]. Mucin stock solution pH was measured using a HI 2211 pH/ORP meter (Hanna Instruments, Leighton Buzzard, UK). The Type II porcine gastric mucin concentrations selected (0.1, 0.3, 0.5% *w*/*v*) represent MUC5A/MUC5B concentrations in healthy individuals (0.1% *w*/*v*) and hypersecretions (0.3, 0.5% *w*/*v*) associated with smokers and individuals with diseases such as asthma, COPD, and cystic fibrosis [21]. A larger droplet size was used to maximise the virus number per droplet (initial radius of 31 ± 1 µm), with five-minute levitations at 40% RH.

### 2.4. Bulk Phase Viral Viability Measurements 

To test the effects of pH on viral infectivity, bulk phase infectivity assays were used to control the pH of the solution the MHV virus particles were subjected to. MHV working stocks were diluted to 10^−4^ into basic DMEM, 10% FBS (pH 7.5, 8, 8.5, 9.5, 10). After a 10-min incubation period, within a sealed tube at room temperature, the samples were diluted back into 5 mL DMEM pH 7.5 to achieve a viral concentration of ~10^−6.5^; this final dilution was transferred evenly across a 96-well plate containing MHV target 17-CL1 cells. Viral concentrations were predicted from the initial working stock titre and serial dilutions were adapted to achieve ~50% of wells being infected for accurate viral quantification. Initial virus infectivity was calculated from the shortest possible incubation, ≤15 s, and this served as a T = 0 to standardise longer incubations (T = 0 + n). There was no difference in bulk phase pH in solutions of 2% and 10% FBS, Appendix A.

### 2.5. Viral Stability Statistics 

To compare the effect of two different conditions on viral viability, an F-Test was used to test the null hypothesis that the variances of two populations are equal, if F > F Critical one-tail, the null hypothesis was rejected. Once the variance was determined, a two tailed, a *t*-Test was used to determine if the variance was significant. 

### 2.6. Comparative Kinetic Electrodynamic Balance Measurements

CKEDB measurements were used to investigate bioaerosol evaporation kinetics and particle phase change. Contrary to CELEBS, a single aerosol droplet of reproducible size (initial droplet radius of 25 ± 1 µm) was generated by a DoD (note, the same type used in the CELEBS instrument) and injected into an electrodynamic balance where it was levitated under environmental conditions measured continuously by RH and temperature probes. The selected RHs and temperature were the same as those studied in the CELEBS infectivity measurements. Each trapped droplet was allowed to equilibrate in line with protocols set out by Rovelli et al. [34]. Captured droplets were illuminated with a 532 nm laser, allowing particle radius and phase approximation to be determined from the scattered light in line with Haddrell et al. [39]. Images were collected every 10 ms from a CCD camera with a central scattering angle 45° allowing accurate phase function measurement from an angular range of 32° to 58°. Droplets compositions were the same as those used in the CELEBS measurements (i.e., DMEM 10% FBS and mucin 0, 0.1, 0.3 and 0.5% *w*/*v*) with measurements made on 100 droplets each for the 5 s period of droplet drying kinetics. Through an analysis of light scattering and particle radius, the time point at which the droplet scatters light consistent with a non-spherical or crystalline particle was identified. Further, the time point at which heterogeneity could be identified in the evaporating droplet was established from a change in scattered light intensity pattern consistent with spontaneous inclusion formation.

### 2.7. Falling Droplet Column

Aerosol droplets of a reproducible size were generated using a DoD (the same type used in CELEBS and CKEDB measurements). A continuous stream of uniform droplets was injected into a vertical glass column of 4 cm^2^ cross sectional area and 50 cm length. RH and temperature were maintained as the droplet fell vertically down the column with aerosol size and morphology measured in line with Hardy et al. [40]. Droplets (of composition DMEM, 10% FBS, and 0% or 0.5% *w*/*v* mucin) were subjected to 40% RH and 19 °C and the dried particles were collected on a glass slide at the bottom of the column for scanning electron microscopy (SEM) analysis. 

Prior to measurement, samples were coated in graphite. SEM images were obtained using JSM-IT300 (JEOL) at room temperature with an accelerating voltage of 15 kV and a working distance of 9–11 mm. The secondary electron detector (SED) provided topographical information capturing the inelastically scattered electrons from the surface. The backscattered electron detector (BED-C) provided particle phase information capturing the elastically scattered electrons. Sample composition can be inferred from the BED-C images with the scattering proportional to atomic number. 

## 3. Results

### 3.1. Viral Infectivity Measurements

#### 3.1.1. MHV Is an Appropriate Surrogate for SARS-CoV-2

Aerosol phase infectivity measurements were used to establish the RH-time dependent loss of infectivity for comparison with that observed for SARS-CoV-2 [12]. At 90% RH, Figure 2a, there was gradual decay in mean virus infectivity, in contrast with the loss of infectivity seen at 40% RH, Figure 2b. To compare the pH-time dependent decay in infectivity, bulk phase infectivity measurements were performed across a range of high pH values. At increased pH, there was a decrease in viral infectivity compared with that measured at pH 7.5 (T = 0), Figure 2c. Equally, the increased pH had a time-dependent effect on virus infectivity in the bulk phase, Figure 2d. In summary, the data suggest a decrease in mean viral infectivity over time that occurs more rapidly at high pH and is similar to that observed for SARS-CoV-2. 

#### 3.1.2. Comparing MHV Infectivity in Droplets of Different Size

Through increasing the pulse voltage and pulse width applied to the DoD, a 1.8-fold increase in initial bioaerosol volume contributed to a 3.5-fold increase in infectious units per droplet, Figure 3a. The data suggest that these two initial volumes show no significant difference in the temporal change in the virus infectivity at 40% RH over five minutes. Regardless of initial droplet volume, a similar loss of infectivity within the first 30 s was observed, Figure 3b.

#### 3.1.3. Mucin Sustains Aerosolised MHV Infectivity 

Virus in droplets containing 0.1% *w*/*v* mucin showed a higher mean infectivity after 30 s levitation compared with 0% mucin (40% RH). Equally, there was a higher mean infectivity in droplets containing 0.3% *w*/*v* and 0.5% *w*/*v* mucin, Figure 4a. This relationship between viral infectivity and mucin concentration continued after 2 min, albeit to a slightly lesser extent, Figure 4b. In contrast, there was no observable effect of mucin concentration on mean virus infectivity after 5 min levitation, Figure 4c. To investigate if the mitigation of infectivity loss by mucin is unique to the aerosol phase, bulk phase infectivity assays were carried out at a concentration of 0.5% *w*/*v* mucin. There was no observable effect of mucin on bulk phase viral infectivity, Appendix A.

In summary, increased concentrations of mucin result in a transient mitigation of viral decay, with significantly higher sustained viral infectivity within the first 2 min of bioaerosol generation. However, this higher infectivity is not apparent at a longer time scale of 5 min for the three mucin concentrations investigated in this study. 

To explore mechanistically how the presence of mucin affects viral infectivity, we now report the effect of mucin on aerosol phase dynamics.

### 3.2. Aerosol Phase Evaporation Dynamics 

#### Mucin Promotes Heterogenous Bioaerosol Phase Change

The representative evaporation profile of a single levitated droplet (DMEM 10% FBS, 0% *w*/*v* mucin) had two distinct regimes, Figure 5a. During the initial evaporation period, the calculated radius decreased, and the scattered light profile was consistent with a homogenous spherical droplet. Further into the evaporation, a second time period is characterised by an irregular light scattering pattern. The distinct fringes seen on the phase function for this period are lost, characteristic of an amorphous or crystalline non-spherical structure. 

Similarly, in a representative evaporation profile for a levitated droplet (DMEM 10% FBS, 0.5% *w*/*v* mucin), there were two distinct regimes during the evaporation. However, there was a change in the scattering pattern towards the end of the initial homogenous evaporation period consistent with the formation of a sphere containing inclusions, Figure 5b. 

There was no significant difference in efflorescence time for DMEM droplets without or with mucin (0.1, 0.3 and 0.5% *w*/*v*), Figure 6a. However, there was evidence for spontaneous inclusion formation dependency on mucin concentration. At lower mucin concentrations, few aerosol droplets showed any phase change before crystallisation with inclusions observed in only a small proportion of the droplets containing 0% and 0.1% *w*/*v* mucin. In contrast with increased mucin concentrations, there was an increased proportion of aerosol droplets which showed the formation of inclusions before crystallisation, Figure 6b. There was no difference in final particle size after evaporation despite mucin concentration, Appendix A. 

In summary, the increased concentration of mucin in the evaporating aerosol resulted in an increased formation of particles containing heterogeneous inclusions during evaporation, and these morphologies were much less common in droplets without mucin. Increased mucin concentrations also shortened the mean time for such morphologies to form and resulted in a more heterogenous distribution.

To visualise the influence of mucin on the morphology of dried particles, particles were collected from the bottom of the falling droplet column and imaged using SEM. The particles dried at 40% RH appear heterogenous in morphology, Appendix A. There is evidence of phase separation with BED-C images showing a contrast between higher atomic number inorganic structures appearing lighter against the darker regions of low atomic number organic matter. SED images indicate the amorphous shape with BED-C images showing the heterogenous inclusion formation below the particle surface, Figure 7.

## 4. Discussion

Presented here is a combined approach to study the biological and microphysical factors influencing airborne virus transmission, incorporating MHV infectivity measurements with bioaerosol evaporation kinetics to interpret the mechanistic drivers behind coronavirus decay in respiratory secretions. In the aerosolised infectivity assays, there was a decay in infectivity within the first 30 s of droplet generation at 40% RH, compared with a more gradual decay in airborne infectivity at 90% RH, both consistent with the observations for SARS-CoV-2 and MHV reported by Oswin et al. [5,12]. To investigate the gradual loss of infectivity at high RH, above the low RHs required for efflorescence [6,12], bulk phase infectivity measurements were performed across a range of higher pH values. These values were chosen to simulate an increase in pH that would occur during droplet evaporation. These combined experiments indicate two mechanisms behind viral decay, a droplet phase change associated with efflorescence, and increases in pH of the evaporating droplet. Although it is difficult to determine airborne respiratory bioaerosol composition, it is understood that bioaerosol pH increases and is heterogenous throughout the droplet [41]. CO_2_ partitioning from the bioaerosol [11] is therefore driving non-physiological conditions within the evaporating droplet and a loss of infectivity. These comparable decay trends for two coronaviruses give confidence that MHV is an appropriate surrogate for SARS-CoV-2 in the aerosol phase.

Noticeably, a 1.8-fold increase in droplet volume resulted in no temporal difference to the loss of infectivity at 40% RH. Restricted by the size limits of the dispenser, the droplets in this investigation (25–31 µm radius) are representative of those respiratory droplets secreted orally through events such as speaking and coughing [9,42,43]. Droplets of this size can travel >2 m, especially in dry environmental conditions [6]. A dried particle could remain airborne for the timescales the virus remained infectious in this study. There is scope to investigate a wider range of droplet sizes to determine viral infectivity dependency on particle volume, with droplets representative of other modes of generations such as breathing. However, our aim is to investigate the microphysical mechanistic factors that impact on airborne survival, factors that can provide information on survival across the wide-size range alongside size-dependent models of the microphysical processes.

There are a multitude of factors that promote or limit airborne transmission between hosts. These factors include, but are not limited to; environmental conditions, viral load, and host intervention strategies [44,45]. Investigation of droplet composition through the introduction of mucin to the bioaerosol saw a transient effect on MHV infectivity, mitigating the viral decay at 40% RH over 2 min. It is difficult to say how this will translate to a viral viability from healthy asymptomatic patients (0.1% *w*/*v* mucin). However, our findings indicate that increased mucin concentrations as a result of respiratory infection may promote viral longevity, temporarily increasing chances of airborne transmission. The CKEDB measurements of consistent evaporation rates between the aerosol of differing mucin concentrations indicate that mucin does not play a significant role in aerosol evaporation kinetics. Instead, mucin spontaneously enhances the formation of inclusions within the droplet which appear to offer protection from deactivation. The viral particle may bind to the inclusions formed during the evaporation of the droplet, thereby shielding the virus from deactivation through the non-physiological conditions inside the equilibrated droplet or particle.

Computational modelling from Dommer et al. [16] supports a protective mucin hypothesis indicating that mucins dispersed throughout the bioaerosol could coat SARS-CoV-2 virions, binding to the charged spike protein (S-protein) through interactions with glycans and calcium ions, thereby molecularly shielding the virus from inactivation during bioaerosol droplet evaporation. The physicochemical properties of mucus are complex and are dictated by hydration and glycosylation of the protein mucin fibres [46]. Non-physiological pH within the droplet could lead to an increase in interactions between mucin and Ca^2+^ ions, resulting in the formation of larger polymeric structures [47,48]. Semi-solid droplet composition has been previously reported for basal cell media containing mucin and calcium through probing the stepwise equilibration state of electrodynamically levitated aerosol, indicating a potential for internal droplet physiochemistry to influence viral infectivity [8].

MHV enters target cells through a non-sialic acid target mouse carcinoembryonic antigen-related cell adhesion molecule 1 (CEACAM1) [49,50]. Some MHV strains express haemagglutinin esterase protein, which does not preferentially bind 5-O-acetylated and 9-O-acetylated sialic acid present in porcine mucin but instead binds 4-O-acetyled sialic acid [51,52,53]. By using a non-physiological system, i.e., MHV and porcine mucin, this research disentangles the direct specific inhibitory binding effects of mucin from non-specific or electrostatic interactions. This is in contrast with bulk phase measurements from Wardzala et al. [54], where they show inhibition of cell infection by coronaviruses through specific interactions between spike proteins and sialic acid residues on mucin. By isolating any specific binding effects and varying the concentration of mucin in the bioaerosol sample, we have been able to deliberately influence the viral infectivity indirectly through the formation of protective mucin inclusions.

This research highlights the limitations of using growth media as a sole aerosolisation ingredient and begins to address these limitations, demonstrating a role for mucin in the aerosol phase. Here, harvested virus from infected cells is used as a working MHV stock to aerosolise droplets in the CELEBS. The main component, DMEM basal cell media [55], is designed to mimic biological conditions at equilibrium for cell culture viability and is often used in aerosol viability studies [12,56]. The complete DMEM formulation includes, 10% FBS with a total estimated albumin concentration of 2.5 g/L, slightly higher than the upper bound of clinical saliva albumin content 0.2–1.6 g/L [56,57]. In comparison with similar studies, there is no indication of the FBS significantly impacting the loss of infectivity [5], or the evaporation dynamics at low RH, with droplets of 2% FBS evaporating within 5 s at 51% RH [12]. In terms of total water content, inorganic salt concentration, and amino acid concentration, basal cell medium is a reasonable model representative of saliva [58,59,60]. Through investigation of a model secretory mucin on infectivity, this work has built a framework on which to investigate a range of artificial and clinical respiratory secretions. Ultimately, this allows for predictions of airborne viral stability based on bioaerosol composition.

The CELEBS measurements directly coupled airborne viral infectivity measurements with aerosol evaporation phase change dynamics on a high time resolution, allowing for the identification of acute phase change within seconds of droplet generation. This unique feature allows for accurate reporting of relationships between aerosol dynamics and viral infectivity. In conclusion, this research has confirmed the role of RH and phase change on the evaporation kinetics of model respiratory secretions and has established the influence of mucin on internal bioaerosol microphysics. Considered as an ensemble, these factors determine the viability of coronavirus in the aerosol phase.

## Figures and Tables

**Figure 1 viruses-14-01856-f001:**
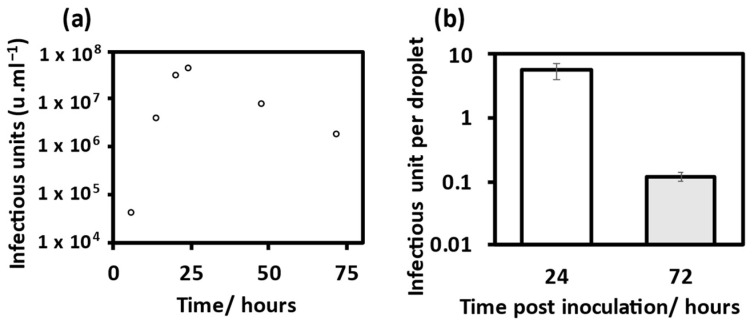
Propagation and TCID_50_ of mouse hepatitis virus (MHV). (**a**) Infectious titre of stock solution retrieved from 17-CL1 cells post inoculation with MHV after incubation period, calculated using TCID_50_. (**b**) Mean infectious unit per droplet for two virus titres from MHV propagation. 24 h inoculation (M = 5.59, SD = 1.57), compared with 72 h inoculation (M = 0.11, SD = 0.017), error bars show standard deviation.

**Figure 2 viruses-14-01856-f002:**
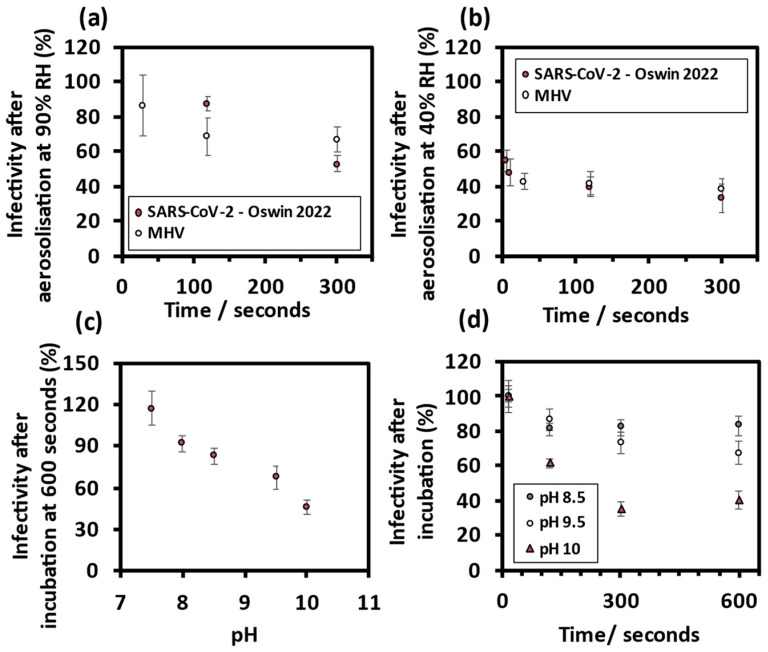
MHV is an appropriate surrogate for severe acute respiratory syndrome coronavirus 2 (SARS-CoV-2). Collected virus infectivity after five minutes aerosolisation, comparing MHV with SARS-CoV-2 from Oswin et al. 2022 at (**a**) 90% RH and (**b**) 40% RH. (**c**) Mean virus infectivity after a ten-minute incubation in high pH DMEM (10% FBS). (**d**) Mean virus infectivity decreases over time with increases in pH of the bulk liquid solutions.

**Figure 3 viruses-14-01856-f003:**
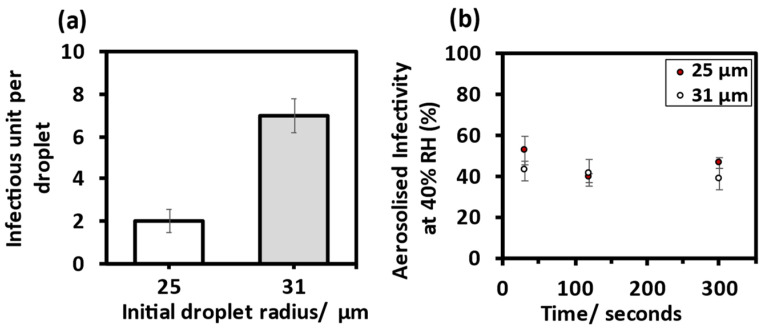
Comparing MHV infectivity in two droplet sizes. (**a**) Mean infectious unit per droplet for initial droplet radii at 90% RH and after 5 s levitations, 25 ± 1 µm (M = 2.01, SD = 1.54) and 31 ± 1 µm, (M = 6.97, SD = 2.46). (**b**) Collected virus infectivity over five minutes at 40% RH, for initial droplet radii 25 ± 1 µm and 31 ± 1 µm.

**Figure 4 viruses-14-01856-f004:**
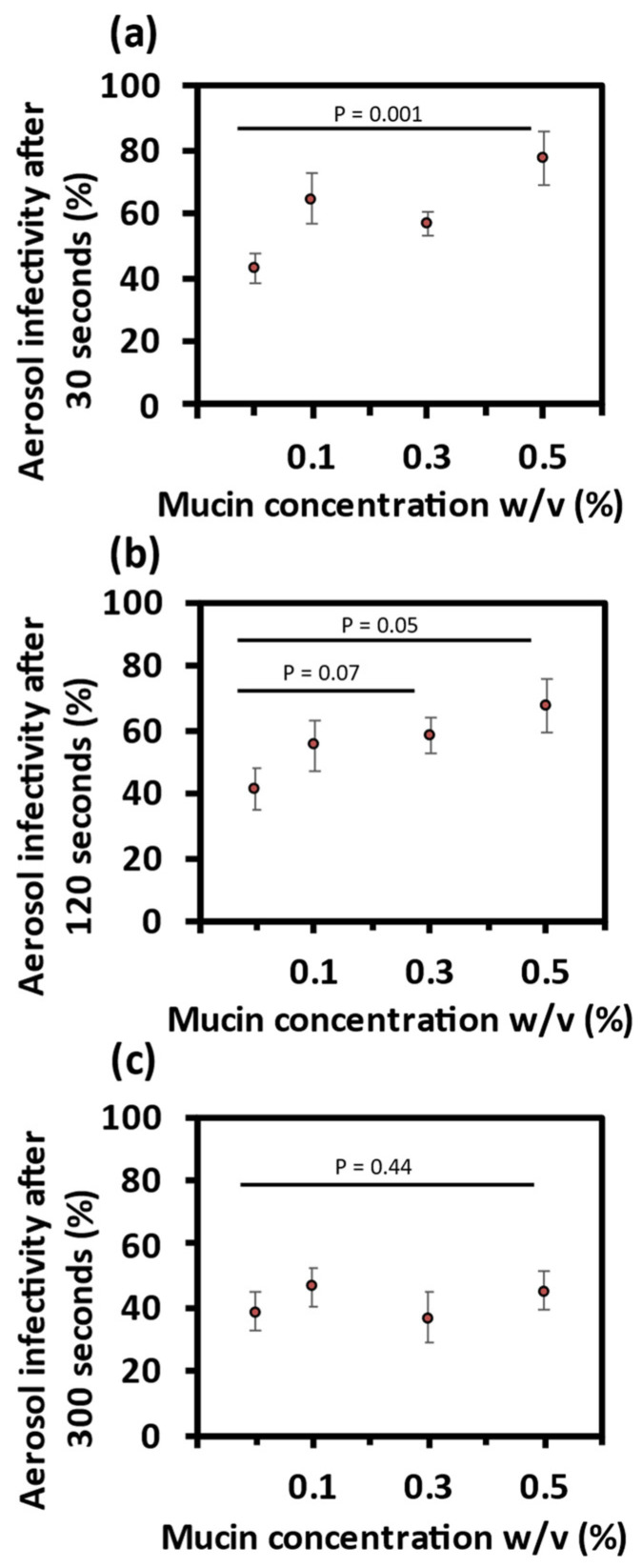
Effect of mucin on airborne MHV infectivity. Collected virus infectivity after levitation at 40% RH with increasing concentrations of mucin. (**a**) 30 s. (**b**) 120 s. (**c**) 300 s.

**Figure 5 viruses-14-01856-f005:**
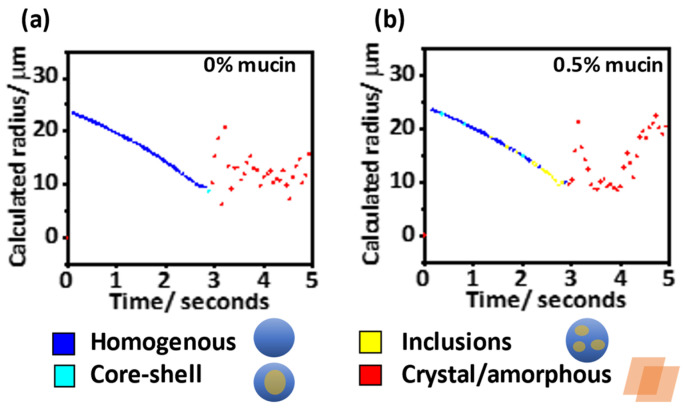
Evaporation profile of representative droplets examining the change in radius and phase morphology over five seconds with a time resolution of 60 ms. Due to chaotic scattering of light from crystal/amorphous particles, radius estimations at later time (red points) are unreliable. Evaporation of droplets containing (**a**) DMEM, 10% FBS and 0% mucin, and (**b**) DMEM, 10% FBS with 0.5% *w*/*v* mucin.

**Figure 6 viruses-14-01856-f006:**
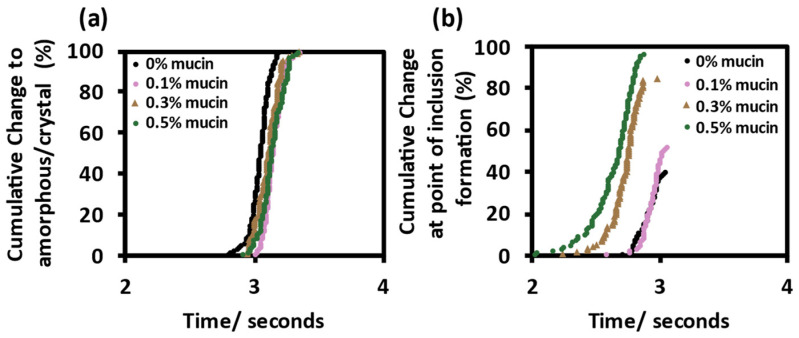
Change in morphology of evaporating droplets. Phase change analysis of evaporating droplets, time resolution of 10 ms, N = 100 (DMEM, 10% FBS and % *w*/*v* mucin). Cumulative change in morphology to (**a**) amorphous/crystal and (**b**) inclusions.

**Figure 7 viruses-14-01856-f007:**
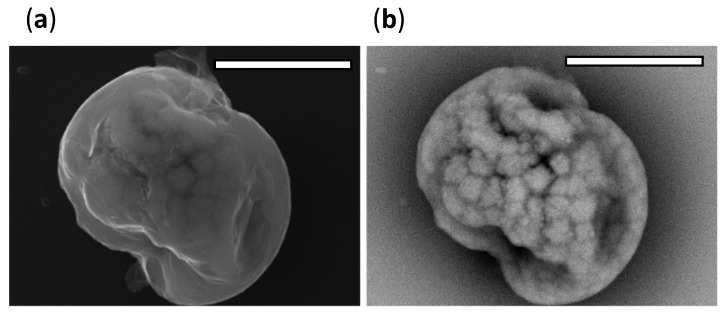
SEM micrographs of a representative particle (DMEM, 10% FBS, 0.5% *w*/*v* mucin) collected from the bottom of the falling droplet column, magnification ×10,000, scale bar 5 µm. (**a**) SED images show topographic information of the particle. (**b**) BED-C images show the presence of inclusions within the dried particle.

## Data Availability

Not applicable.

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
