# Peer review of "Mucin Transiently Sustains Coronavirus Infectivity through Heterogenous Changes in Phase Morphology of Evaporating Aerosol"

_viruses, 2022, doi:10.3390/v14091856_

Round 1

Reviewer 1 Report

The manuscript of Alexander et al reports the use of sophisticated biophysical analyses to assess the effect of mucins on coronavirus infectivity in aerosols. As such, it could serve as a useful addition to the knowledge base of the coronavirus pandemic, and interactions between mucins and viruses in general. However, it is marred by several issues at present.

The most substantial is the discussions of the mucins used in the study and the expression of mucins in vivo. On line 84 the authors state that MUC2 is a component of stimulated saliva, but the major secreted mucins in saliva are MUC5B and MUC7, so this represents a major error because it serves as a lead-in to the experimental methodology. In Methods on line 132, the authors should specify which mucins are predominantly present in Sigma-Aldrich porcine stomach mucin Type II. The predominant secreted mucins of the stomach are MUC5AC and MUC6, but whether the commercial product reflects that should be confirmed by the manufacturer and/or published literature because it is a key reagent throughout the studies. In Results, line 267, the mucin molecular species in Type II mucus should be specified.

 On line 73, the authors state that mucus provides a source of nutrition for the airway microbiota; while this is certainly true in the intestine, and may be true in the mouth, it is almost certainly not true in healthy intrapulmonary airways. In that location, the microbiota probably consists mostly or exclusively of organisms aspirated from the oropharynx, which are present only transiently because of ciliary clearance of the mucus layer. In static airway mucus in diseases such as cystic fibrosis and obstructive bronchiolitis, mucus may provide a source of microbial nutrients, but the authors need to be more clear and accurate.

 The Methods should include a section on EM methodology, and a short general section on statistics. Regarding the latter, it would be reasonable to state in the methods and the first figure legend that data show means and standard deviation throughout, and leave that out of subsequent figures. However, some figures show standard error, and an explanation of why each analysis (SD vs SEM) is used in each figure should be provided in the statistical section if it does vary, otherwise the analysis should be made uniform. It would be reasonable in the figure legends to write “N = 4” rather than writing out “Data are the means of 4 measurements”.

 The figures should be revised to include more conventional axis labels. For example, in Figure 2, the Y-axis is labeled “Infectivity/%”, which this reviewer has never before seen. A more conventional label would be “Infectivity (%)”. Also, some of the figure legends contain odd and unnecessary words such as line 229 “data from this research at of MHV”. Figure 7 could be made more intuitive for the reader if the concentrations of mucins were labeled on the left side of the panels, and the type of EM imaging labeled on top (were both SEM or only the left column?). Also, is it correct that both (b) and (c) used 0.5% mucin, and if so, why the difference in appearance?

 In general, the manscript is well written. However, there are several opportunities to improve phrasing. For example line 26 should be “effect” rather than “affect”. Line 74 should be “microbiota” rather than “microbiome”. Line 162 should be “Aerosols”. Line 195 should be “bioaerosols”. Line 253 seems to be missing a word, perhaps “infectivity from droplet”. Line 279, “all be it” should instead be “albeit”. Line 405 should be “an appropriate”. Reference 16 is a bioRxiv preprint and should be updated if possible.

Reviewer 2 Report

Review of manuscript ”Mucin transiently mitigates the loss of coronavirus infectivity in aerosol through interruption of particle efflorescence”.

This study about the effect of mucin on droplet drying process is interesting and contributes with another important piece of information to our understanding of aerosol transmission of viral infections. Droplet microphysics is likely central to airborne virus infectivity, but so far, many studies have investigated this with rather blunt methods. The methodology in this study has a high potential and the investigation of mucin is novel. However, the way the study is presented in text and figures does not impress. The new contribution of this study can be condensed considerably – I would suggest to half the amount of text. Paragraphs are too long, sentences are too long and words are too long. Your results are hidden in too much text making it extremely difficult for the reader to grasp your findings. Better structure of methods and results are needed. I would also recommend proof reading more carefully.

Please remove 99% of all “there is evidence” and “there is clear evidence”. Especially from the result section. If there is evidence, make the reader understand what your results show. Simply writing that there is evidence will not convince anyone (rather the opposite). Statistical tests can be used to show where there is a statistically significant difference.

Methods

The same statistical measures and tests are used throughout the study, thus, I would recommend having a statistics section in the methodology for the explanation, and then simply refer to the name of the test in the result section. When it comes to the number of repeated measurements, you write that “data are means of x measurements”. Why the bigger than? Are data in the same figure repeated different amount of times? The number of measurements varies between ≥3 to ≥6, is there a reason for this? What is “one measurement”?

The use of the word “bioaerosols” is not correct. One bioaerosol is not one of your levitated droplets/particles. I would suggest using droplet or particle for this.

I appreciate that you clearly write what is bulk measurements and what is aerosol measurements. However, the bulk measurements are not well defined. What is the volume of bulk experiments? What container and what environment was the experiment performed in? What does “rapidly diluted back into neutral bulk phase” imply?

Results

From the method and the title, my expectations on what to read under the result heading is what comes at row 274. The first paragraph, 3.1, I would put under methods, considering the scope of the article.

3.2: It is difficult to follow the text. Since all the data is in the figure, could you condensate what is actually essential?

The result section is full of discussion sentences that should be removed since you have a separate discussion section. If you want to have a separate result section, please keep this plain with description of your findings without assessments, comparisons and conclusions; for example, sentences that start on rows 196, 205, 216, 220, 253.

Figure 2: Put more information in the figure, not only in the figure text. This goes for all figures. Indicate that the upper two are aerosol phase and the lower are bulk phase and that the SARS-CoV-2 data is from another study. Why do you have measurements on pH 7.5, it says on 7 in the method? For the discussion, could you add a comment on why the infectivity increased to 120% at pH 7.5?

Row 237-242 is pure methodology. Remove from results.

Row 253: (preferably moved to discussion) Your whole method and scope is that evaporation and efflorescence causes infectivity loss, but here you state that it has nothing to do with the bulk to surface ratio. If you really want to write this, you need to build up your argument better and compare for example efflorescence times for the different droplet sizes. In my eyes, 25 and 31 µm is on the same scale and I would not make that statement based on this small difference. If you had compared 0.5 µm with 2, 10 and 31 µm, then you have investigated this question. Now you just have a loose statement saying that there is little difference between 25 and 31 µm. How precise is you DoD by the way?

Row 265-273: Move to methodology. Be consistent in writing type II or type 2 mucin.

It is extremely challenging to follow all the different experiments you have done and you spend quite some text on describing the methodology next to the results (for clarity). Could you help the reader with a table of all droplet sizes, RH’s, levitation times, bulk volumes, pH and other things and where they were used? And define what “one measurement” is. Then you could perhaps name the settings (A-1 for aerosol test 1) and give it short in parenthesis where you present the results. Then you could also skip all the “in order to evaluate...” sentences having this in the table and just present results under results.

Discussion

One major thing that is missing in the discussion is to put your results into the perspectives of reality. You spend a lot of text on highlighting the CELEBS-CKEDB method and less on the actual findings and how results compare to previous literature. Can you explain or at least discuss why MHV infectivity changes with the inclusions that you find? Are there any previous hypotheses? How does it compare to the findings in references 7 and 8 where efflorescence had protective effects?

You write that the low mucin concentration of 0.1% w/v represents healthy (likely also pre-and a-symptomatic) subjects but at this low level you see no significant change from 0%. So are COPD patients and CF patients emitting more infectious viruses? For Covid, the major part of disease transmission occurred from persons yet not experiencing symptoms. Your results are only significant for high concentrations and on a short, ~2 min, time scale. Please address the relevance of these conditions for aerosol transmission. In the discussion (row 417) you write that mucin had a distinct effect on MHV infectivity, here you should add “high concentrations of mucine” as there is no significance to the “normal” concentration 0.1 %.

What is the effect of the added 10% FBS? FBS contains proteins and other stuff that could influence the droplet chemistry. You write that DMEM is representable of saliva, but is that the case also when adding 10% FBS? The FBS changes properties as surface tension, viscosity and density.

Respiratory droplets are generated from several regions of the respiratory tract. Your liquid and droplet sizes primarily represent oral aerosol emissions. Can you comment on the generalizability for respiratory aerosols that contribute to disease transmission?

The last paragraph before conclusion feels odd compared to what the rest of the text is about. Could this be integrated in a better way?

Title

From my understanding of your results, mucin do not interrupt particle efflorescence, as you show in figures 5 and 6. You have chosen formulations as “changed phase morphology” and “inclusion morphology” in the text, but not interruption. It would be more correct to include that this is only for short time scales of ~2 min, and not an overall effect.

Sentences that are four rows long, where the reader cannot follow the meaning: row 46-49, row 82-85, row 85-89, row 114-117, row 178-182, row 230-233. Please revise.

Minor comments:

Row 124: Poisson distribution

Row 129 (and other places): Did you have temperature control at the level of two decimals?

Row 131: Is the mucine or the porcine stomach of type II?

Row 132: Sigma-Aldrich? Be consistent writing country or not.

Row 149: were (not are, keep to one tempus)

Row 149 and 164: what does “tight control” mean?

Row 154: According to the method, you have 10 ms resolution in the CCD camera, but in figure 5 you have data points representing binned averages of 5 ms time points?

Row 156: Reformulate the sentence starting with The evaporation kinetics.

Row 162: Droplets of reproducible size…

Row 164: Cross-sectional area?

Row 167: What is meant by dilution of aerosol? Dilution of the droplet number? Dilution of liquid? If it is the same DoD as in CKEDB, why do you need to change to ensure reproducibility?

Row 170 (and other places): “evaporated droplets” sounds like they do not exist anymore (or only in gas phase). Consider changing to dried particle or similar

Row 178: What is the definition of “short five seconds”, are they shorter than long five seconds?

Row 186: Infectious titre of stock solution (not of the particles)

Figure 1a: Unit on y-axis is confusing

Row 214: Are not

Row 216: What is the “neutral” pH of the DMEM that you use? (you dilute back to neutral but state that there was no difference at pH 7 so is 7 the reference?)

Row 226-233: Revise the figure text. “Mean virus infectivity over time at a pulse width of 120 µs”? The reader do not know the implications of a 120 µs pulse width. “data from this research at of MHV 40%”? “Mean virus infectivity over time bulk phase reactions data”?

Row 243: 2 infectious units per droplet compared to 7, is that really 2.5? Can you also add if there was a significant increase per droplet volume?

Row 288-294: excess text that you just presented and have in the figure, remove

Figure 4: What is the x-axis value for the leftmost data points? 0? It is also more common to have time on the x-axis. Could you make one big graph for all three data series?

Row 299: Mean virus infectivity after a) 30 s, b) 120 s, and c) 300 s.

Row 306: Add 10% FBS after DMEM

Row 326: “a single bioaerosol model 40% RH”? Is that one levitated droplet? At 40% RH? Was this droplet representative of other droplets?

Figure 6: Change colors so that one can see the difference. Increase font size on legend text. Write headings explaining what the graphs represent. If you have 100 droplets, maybe the y-axis could have 100 as maximum?

Row 356: 100 measurements or 100 droplets? Is there a difference?

Row 366 “evaporated particles” – do they exist?

Row 368: How do you analyze what the light and dark regions consist of?

Row 373: “the presence of efflorescence..” efflorescence cannot be present, it happens or not

Figure 7: Indicate scale bar length in the pictures. Clarify in the figure what the difference is between the left and right column of pictures. Considering the scale bar length being fairly similar in b and c, it does not seem like there is a 10x difference in magnification. How representative are these dried droplets? Did you look at 100 similar droplets or were these unique?

Row 380: “..after falling at 40% RH” sounds weird.

Row 393: “Used as a common surrogate in respiratory aerosol formulation..” I do not see what this refers to without a reference. What is respiratory aerosol formulation? Cell media is common when studying bioaerosols in laboratory settings.

Row 405: an appropriate

Row 407: please define “near instantaneous”?

Row 410: “..to higher pH” – at higher pH?

Round 2

Reviewer 2 Report

I appreciate the changes made in accordance with my previous comments – the manuscript has improved a lot. However, I still have some concerns regarding this manuscript. There are some sentences and expressions that I think reflect carelessness in the writing and editing procedure, and I find it hard to believe that all authors have read this manuscript in detail. To give some examples, I have never heard of “cytopathic death” and it gives 81 hits on Google Scholar – meaning that you either have to explain this quite uncommon term, or was it “cytopathic effect” that you investigated, as most studies in this field do? “Efflorescence event” is another low-hit search term that I would consider a physical process rather than an event. In a sentence added in the discussion, you state that your DoD can generate droplets in the size range 50-75 µm, while your results show that you studied 25 and 31 µm droplet radius – choose one way of presenting the droplet size, either diameter (the most commonly used) or radius. The y-axis labels in Figure 2a “aerosolized infectivity..” – what you measure is the MHV infectivity after collection, not the aerosolized infectivity. The legends in Figure 2 still requires a magnifying glass. The sentence: “…promote airborne or limit propagation between hosts..”? – propagation normally takes place within a host. There are missing words and commas in some places, please have the text checked by a professional.

Then I would like to address another thing. Why is the N, the number of repeated measurements, now very nicely presented in the supplementary table, varying from 3 to 22? It seems you get a lower SD with lower N, so it looks like it affects your comparisons. I have full understanding of that biology and virology have sometimes quite big variations within a measurement set, but I do not see the reason for your big variation in N. It is also common to use non-parametric statistical tests when N is low or not similar between groups, why did you chose parametric test? In addition, you now explained that one measurement is one single population of droplets, but how big is a single population?
